IMPACT: an interactive multi-disease prevention and counterfactual treatment system using explainable AI and a multimodal LLM

http://orcid.org/0000-0001-5947-2043 Mohanty Prasant Kumar 1
Anand John Francis Sharmila 2
Barik Rabindra Kumar 3
http://orcid.org/0000-0003-2492-3312 Reddy K. Hemant Kumar 4
Sinha Roy Diptendu 1
http://orcid.org/0000-0001-6656-4333 Saikia Manob Jyoti 5 6 msaikia@memphis.edu
1 Department of Computer Science and Engineering, National Institute of Technology , Shillong, Meghalaya , India
2 Department of Computer Science, Rijal Alma’a, King Khalid University , Abha , Saudi Arabia
3 School of Computer Applications, KIIT Deemed to be University , Bhubaneswar, Odisha , India
4 Department of Computer Science and Engineering, VIT-AP University , Beside AP Secretariat Amaravati, Andhra Pradesh , India
5 Biomedical Sensors & Systems Lab, The University of Memphis , Memphis, TN , United States
6 Electrical and Computer Engineering Department, The University of Memphis , Memphis, TN , United States
Lovino Marta
Electronic publication date: 2025 Apr 29
Publication date: 2025
Volume: 11
Electronic Location ID: e2839
Received 2024 Nov 7; Accepted 2025 Mar 26
Copyright: © 2025 Mohanty et al.
Copyright year: 2025
Copyright holder: Mohanty et al.
License: This is an open access article distributed under the terms of the Creative Commons Attribution License, which permits unrestricted use, distribution, reproduction and adaptation in any medium and for any purpose provided that it is properly attributed. For attribution, the original author(s), title, publication source (PeerJ Computer Science) and either DOI or URL of the article must be cited.
License URL: https://creativecommons.org/licenses/by/4.0/

Keywords: LLM, Explainable AI, Large language model, NSGA-II, Healthcare, Treatment

Funding: Biomedical Sensors & Systems Lab at the University of Memphis, Memphis, TN, United States This research and article processing charges were funded by the Biomedical Sensors & Systems Lab at the University of Memphis, Memphis, TN, United States. The funder had a role in study design, data collection and analysis, decision to publish, and preparation of the manuscript.

==============================
Multi-disease conditions strain the body’s defenses, complicating recovery and increasing mortality risk. Therefore, effective concurrent prevention of multiple diseases is essential for mitigating complications and improving overall well-being. Explainable artificial intelligence (XAI) with an advanced multimodal large language model (LLM) can create an interactive system enabling the general public to engage in natural language without any specialized knowledge prerequisites. Counterfactual explanation, an XAI method, offers valuable insights by suggesting adjustments to patient features to minimize disease risks. However, addressing multiple diseases simultaneously poses challenging barriers. This article proposes an interactive multi-disease prevention system that uses Google Gemini Pro, a multimodal LLM, and a non-dominated sorting genetic algorithm, namely NSGA-II, to overcome such problems. This system recommends changes in feature values to concurrently minimize the risk of diseases such as heart attacks and diabetes. The system facilitates personalized feature value selection, significantly reducing disease attack probabilities to as low as possible. Such an approach holds the potential to simultaneously address the unresolved issue of preventing and managing multiple diseases for the general public.

Introduction

Preventing diseases is crucial because it saves an individual from experiencing distress and potential complications, reduces the burden on healthcare systems, and improves overall community health (Kisling & Das, 2023; Caron et al., 2023). When multiple diseases strike an individual simultaneously, the situation becomes drastically precarious and can even lead to a disaster. As the body’s immune system is overwhelmed, the diseases can interact unpredictably, often exacerbating mutual effects. This complicates both treatment and recovery (Budreviciute et al., 2020). The necessity to simultaneously ward off multiple diseases is paramount due to the severe complications that arise when an individual is afflicted by more than one illness at a time. In multi-disease scenarios, attempting to prevent or treat one disease can inadvertently worsen the condition of another, creating a complex and delicate balancing act for healthcare providers (Baccarelli, Dolinoy & Walker, 2023). This interaction can amplify the effects of the diseases, making them more challenging to manage and treat effectively. This complexity is particularly critical in scenarios where healthcare resources are stretched thin, or individuals have underlying health conditions that make them more vulnerable. Hence, understanding and implementing effective disease prevention strategies are paramount in safeguarding public health and maintaining societal stability.

The evolving capabilities of artificial intelligence (AI) and machine learning techniques can significantly enhance effective disease prevention strategies. These technologies can analyze vast amounts of health data, identify patterns that human observers might miss, and predict outbreaks before they become widespread, enabling timely interventions (Olawade et al., 2023). However, the primary limitation of these AI-driven methods lies in their often opaque decision-making processes, which can lead to trust issues among healthcare professionals and patients. This is where XAI (Dwivedi et al., 2023), particularly counterfactual explanations, plays a crucial role. Counterfactual explanations (Del Ser et al., 2024) provide insights into how modifying specific feature values can help prevent a particular disease. However, counterfactual explanations face a major challenge when a patient is at risk of multiple diseases simultaneously: they often generate distinct recommendations for each disease rather than a unified recommendation that accounts for all conditions. In addition, in some cases, these recommendations may be contradictory: modifying a characteristic to reduce the risk of one disease could inadvertently increase the risk of another. Given the growing prevalence of comorbidities, it is essential to develop explainable AI techniques that generate integrated counterfactual recommendations, ensuring that interventions are effective for individual diseases and harmonized for multiple concurrent conditions.

Handling a multi-disease scenario in healthcare analytics can be tackled using a multi-dimensional counterfactual explanation model, which employs a non-dominated sorting algorithm, specifically NSGA-II (Ma et al., 2023). This system aims to generate multiple optimal sets of changes to patient feature values (like lifestyle or biochemical parameters) that can simultaneously minimize the risks of various diseases. Each set of changes is evaluated to ensure it does not inadvertently increase the risk of one disease while decreasing another, promoting a holistic approach to disease prevention. However, despite its efficacy, such a counterfactual explanation model often struggles with broader adoption among the general public, primarily due to its complexity and technical nature, which can be challenging for non-specialists general public to understand. One of the most efficient multimodal large language models, which seamlessly integrates text, image, and video understanding due to its advanced architecture, enabling high-performance tasks across multiple modalities, is Google Gemini Pro (Gemini Team Google et al., 2024). Optimized training techniques and extensive fine-tuning of diverse datasets further enhance its efficiency. Such a model would simplify the interpretations of the counterfactual outputs. Furthermore, the Gemini-enabled model will work simply as a natural language conversation with the system without creating any complexity that the general public hesitates to use, thereby enhancing the practical usability of counterfactual models in preventing multiple diseases.

A proposed Interactive Multi-disease Prevention and Counterfactual Treatment (IMPACT) system represents a significant advancement in delivering customized health advice through a user-friendly digital platform. This innovative approach leverages cutting-edge technology to facilitate more accessible health management tools, particularly for individuals with limited technical expertise. Here are the key contributions of this research: Tailored health recommendations: The model provides personalized health guidance designed to prevent multiple diseases simultaneously, making it highly relevant for individuals seeking comprehensive health management strategies.

Advanced prompt engineering and generative models: Utilizing state-of-the-art prompt engineering and generative models, such as Google Gemini Pro, the system seamlessly integrates user data to generate personalized insights, ensuring the recommendations are accurate and applicable.

Wearable sensor integration: By incorporating data from wearable sensors, the system effortlessly translates real-time health data into actionable advice, democratizing access to health analytics without requiring user proficiency in technology.

Multi-dimensional counterfactual explanation: Employing a sophisticated multi-dimensional counterfactual explanation model guided by the NSGA-II, the system optimizes interventions to minimize the risk of multiple diseases, providing a proactive approach to disease prevention.

User-friendly interface: The interface mimics natural language conversation, making it extremely approachable and easy to use, enhancing user engagement and the overall effectiveness of health recommendations.

This research introduces a novel integration of complex algorithms and user-centric design to create a proactive, personalized health management tool accessible to a broad audience. The codes are made publicly available on: https://doi.org/10.5281/zenodo.14891893.

The organization of this article is as follows: The next section (“Related Works”) briefly describes several approaches for disease prediction or prevention, followed by their gaps. Section “Methodology” describes the procedures adopted, including disease prediction, the proposed IMPACT system and its interactive model, and the multi-dimensional counterfactual model. Section “Experimental Details, Results and Discussions” presents the experimental details and the various results obtained under different scenarios. The complete work is summarized, and future work is outlined in Section “Conclusion”.

Related works

The burgeoning field of AI-driven health analytics has demonstrated significant potential in enhancing disease prevention strategies through advanced data analysis and predictive capabilities. Recent literature extensively explores integrating machine learning techniques, including explainable AI (XAI) and counterfactual explanations, to address complex healthcare challenges, particularly in multi-disease scenarios. This literature review delves into these technologies’ current state and evolution, assessing their impact on healthcare delivery and their inherent difficulties in practical applications.

Artificial intelligence (AI) and machine learning (ML) have revolutionized the healthcare field, particularly in the prevention and management of diseases. Several studies have demonstrated the capability of these technologies to predict disease outbreaks (Heidari et al., 2022), identify risk factors (Ganesh & Kalpana, 2022), and recommend preventive measures with high accuracy. For instance, predictive ML models have been adept at identifying patterns in vast health datasets, leading to early detection of potential health risks before they manifest into more severe conditions (Sang et al., 2024; Zhang et al., 2022). The proposed framework in Tang, Wong & Yu (2023) integrates a Gaussian randomized mechanism for privacy preservation and a two-step domain adaptation method to bridge the domain gap, enhancing the efficiency of multi-site ocular disease recognition through a privacy-preserving federated learning approach. However, despite significant advancements, the application of AI in handling multiple diseases simultaneously remains a challenge, as these models often do not account for the complex interactions between different diseases (Pham et al., 2022). Another challenge is that most of these approaches work towards disease detection, not prevention, and these methods are not trustworthy among people.

The emergence of XAI and counterfactual explanations offers a solution to some of the challenges traditional AI systems pose, especially regarding transparency and trustworthiness. XAI facilitates a better understanding of AI decisions by healthcare providers and patients, thereby increasing their acceptance and implementation (Albahri et al., 2023). Counterfactual explanations, in particular, have shown promise in illustrating how slight changes in lifestyle or other health factors could prevent disease outcomes (Mertes et al., 2022; Wu et al., 2021). Despite these advances, the integration of counterfactual reasoning in multi-disease scenarios is still developing, with existing research primarily focusing on single-disease interventions. This gap underscores the need for models that can effectively handle the nuances of multiple concurrent diseases.

Recent developments in advanced prompt engineering and generative models, like Google Gemini Pro and Chat-GPT, have further enhanced the capability of AI to deliver a better healthcare system. Prompt engineering is a critical technique that involves designing and optimizing prompts to enhance model performance on specific tasks, which has become increasingly important in the healthcare domain (Meskó, 2023). Prompt engineering can help medical professionals in various ways, such as diagnosis, treatment selection, risk assessment, facilitating administrative tasks, improving communication between providers and patients, and assisting in research and education (Meskó, 2023). Google’s Gemini Pro is a competent multi-modal AI model that can analyze complex patient data and medical literature to provide insights that support disease detection and prevention (Saab et al., 2024; Pal & Sankarasubbu, 2024). Prompt engineering can make Gemini Pro more accessible to the general public by creating intuitive interfaces, targeted responses, and addressing ethical considerations (Agrawal, 2024; Balasubramanian & Dakshit, 2024). Despite their sophistication and user-centric design, these models often face challenges in seamlessly integrating complex medical data from various sources, mainly wearable health devices, and converting it into actionable insights for multi-disease prevention without overwhelming the user.

These works of literature reveal a notable gap in the application of AI and ML technologies in the healthcare domain, especially in developing solutions that effectively manage the complexity of multiple disease conditions simultaneously and ensure user-friendly interaction. Our current work aims to bridge this gap by introducing an integrated approach that combines the strengths of AI, XAI, and advanced generative models. By doing so, this work strives to develop a comprehensive health management tool that is effective in disease prevention and accessible and easy to use for a broad audience.

Methodology

The proposed healthcare IMPACT system is designed to offer personalized health recommendations to the general public without requiring technical expertise, presenting information as a simple conversation with the system. Individuals seeking prevention from multiple diseases receive simplified recommendations for various feature values. Following these recommendations can minimize the likelihood of simultaneous disease occurrences. In this healthcare model, the individual acts as an active participant interacting with the system, while multiple disease datasets serve as the other crucial participants, ensuring the model’s functionality.

Data collection

The initial and critical phase necessitates collecting data sets from various hospitals and healthcare centers to develop a recommendation system to prevent multiple diseases. Due to the comprehensive nature of these data sets, categorization into specific disease groups is imperative. Upon organization, these categorized data sets are rendered significantly more viable for analyzing multiple diseases.

Let the number of hospitals be denoted by n. Several disease datasets were collected from each hospital. All hospitals do not need the same number and type of disease datasets. For instance, hospital h1 possesses m disease datasets, hospital h2 has i, and hospital hn has j different disease datasets, with m≠i≠j being a likely scenario. Similarly, the d1 disease dataset from hospital h1 does not have to be identical to the d1 disease dataset from hospital h2 ( Dh1,d1≠Dh2,d1). In Eq. (1), datasets are segregated according to each disease ( Di), where α and β represent two different hospitals. In Eq. (2), Dtot is the set of all the disease datasets, with k representing the total number of diseases.

(1) Di=⋃k=1nDhkdi,whereDhα,d1=Dhβ,d1andα≠β

(2) D_tot={D1,D2,D3,…Di,…,Dk},wherei∈{1,2,…k}.

This recommendation system for preventing multiple diseases is meticulously designed to be highly personalized, adapting seamlessly to each user’s unique characteristics and needs. Initially, the user selects a specific set of diseases from those available in the system’s disease-wise datasets. This selection is designated as the user’s target disease. Personalization is emphasized by allowing different users to have distinct sets of target diseases, addressing their health concerns and risks. Equation (3) defines the target diseases D_tgti for each user Ui, which must be within D_tot.

(3) D_tgti⊆D_tot.

The requirement for personalized data extends to specific user information, which is crucial for each target disease ( D_tgti). Among the features of these target diseases, certain personal information, such as age, gender, weight, job type, residence type, and smoking status, must be provided by the IMPACT system during user interaction, as these features vary among individuals. Furthermore, wearable devices contribute valuable data by monitoring dynamic health indicators such as hypertension (H), blood sugar (BS), and oxygen levels, thereby further enriching the user profile.

The system differentiates between changeable features, such as weight and smoking status, and immutable ones, such as age, gender, and marital status, a distinction crucial for its recommendation mechanism. The system aims to minimize the risk of targeted diseases by proposing optimal values for adjustable features. For example, a suggested body mass index (BMI) target of 20.68153 can effectively lower associated health risks. Users can specify a range for any changeable feature to enhance personalization, such as a BMI range of 22 to 30. The system then recommends a value within this range, like 24.456, optimizing health outcomes within the user’s specified constraints. This method ensures recommendations are scientifically sound, practically achievable, and tailored to individual circumstances.

System model

This healthcare IMPACT system offers a highly personalized approach to disease prevention that adapts to each user’s unique features and preferences. The system model is depicted in Fig. 1. Utilizing disease data sets to predict multiple diseases begins with collecting diverse data ( Master_Dataset(MD)) from various hospitals. This comprehensive data is meticulously organized by disease type, resulting in a structured repository ( D_tot). Users seeking to prevent multiple diseases can select a set of diseases from this repository ( D_tot). This user-specific selection is mathematically defined in Eqs. (3) and (4), where each user, denoted as Ui, has a unique set of targeted disease data ( D_tgti).

(4) D_tgti={D1,D2,…,Dn}.

Figure 1 System model for the proposed IMPACT system.

Sensor data from smartwatches, fitness trackers, and biosensors are collected and integrated into the system, providing physiological and behavioral features such as heart rate, blood pressure, oxygen levels, sleep patterns, and physical activity. Users can also input additional health information through an interactive interface. The collected data undergoes preprocessing to ensure compatibility with the ML framework. Time-series sensor data is processed using statistical feature extraction (mean, standard deviation, and entropy), while discrete and categorical inputs are normalized using Min–Max Scaling and One-Hot Encoding. Missing data are handled through forward filling and KNN-based imputation. These approaches align with best practices in handling missing data in AI-driven models for complex systems, as discussed in Sharifi et al. (2024). The ML model employs a multi-input neural network, where convolutional layers extract temporal patterns while fully connected layers process structured inputs. These feature representations are then used to define the population for the multi-disease prevention problem, ensuring an effective and scalable health prediction system.

A spectrum of sophisticated machine learning models is deployed to ensure accurate predictions of the targeted diseases. The most effective model for each disease is determined through rigorous evaluative metrics, providing the best fit for each disease. Given the variety of diseases, denoted as n, a corresponding number of machine learning models ( n) is required. Each model is tailored to the characteristics of a specific disease dataset. For any user, Ui, the array of models employed can vary, adapting to the unique combination of diseases they are addressing, as represented in Eq. (5). These multiple models define numerous objectives for the multi-disease prevention problem.

(5) Mi={M1,M2,…,Mn}.

Prediction models for all target diseases and user feature information are obtained. Subsequently, a multi-dimensional counterfactual explanation is designed as in Fig. 2, utilizing the sophisticated multi-objective optimization technique known as the Non-dominated Sorting Genetic Algorithm II (NSGA-II) (Kalyanmoy, 2002). It starts by generating an initial population based on feature information. It is followed by iterative processes of calculating multiple objective functions using trained machine learning models, performing non-dominated sorting, and selecting elite populations. The process continues until convergence is reached, leading to optimized and personalized health recommendations, minimizing the possibility of simultaneous attacks of multiple targeted diseases.

Figure 2 Multi-dimensional counterfactual model.

Interactive model

With the help of Google Gemini Pro (as in Fig. 1), the interactive model of the proposed IMPACT system can convert user natural language input to SQL by leveraging its advanced natural language understanding and generation capabilities. The system can accurately interpret intent and context by processing a user’s natural language query and then generating the corresponding SQL query to retrieve the desired data from a database. This lets users interact with databases using natural language, simplifying data access and analysis.

Initially, when a user requests the interface through natural language input, it is processed by the Google Gemini Pro. A second input is required from the prompt, which commands the conversion of the user’s natural language input to SQL. The specific command varies based on the requirement. For instance, if the user intends to retrieve information from the database (in this case, Sqlite3 DB), the command will be converted to a select query. This allows the user to obtain feature details and their values, which are then returned as a response from the model’s interface. In other scenarios, when the user provides feature values, the information is returned as a result of the user data and processed for further operations. The interactive model in Fig. 1 illustrates these interactions between the user and the model.

Similarly, the user interacts with the model using natural language through the interface and gets responses or shares the user data as feature information to the multi-dimensional counterfactual model as it is defined in Algorithm 1. This multi-dimensional model generates the result in the form of personalized health recommendations. Finally, this recommendation is provided to the user to prevent the simultaneous attack of multiple diseases.

Algorithm 1 IMPACT system for feature information.

1: Initialisation: Store the feature information with values for categorical features in the SQLite3 DB database	
2: while user interacts with the interface do	
3:  if task is to retrieve feature information then	
4:   U ← user_input() {Natural language request from the user}	
5:    P ← prompt_input() {Prompt text to convert U to an SQL query}	
6:    Q ← get_gemini_response(U, P) {Google Gemini Pro generates SQL query}	
7:    R ← execute_query(Q) {Execute the SQL query Q on the SQLite3 DB}	
8:    I ← retrieve_info(R) {Retrieve the information I from the database and sent to user}	
9:  end if	
10: if task is to provide feature value then	
11:   V ← user_input() {User provides feature values using natural language}	
12:   DV ← store_values(V) {Store these values V in a data structure DV}	
13:   FI ← FI ∪ DV {Add DV to the feature info. FI}	
14:  end if	
15: end while	
16: RH ← multi_dimentional_counterfactual (FI,Mi) {Generates personalized health recommendations RH using FI and models Mi for each disease Di}	

Multi-dimensional counterfactual model

In Explainable AI for disease prediction, counterfactual explanations fall short when dealing with multiple diseases simultaneously, as they typically focus on a single disease model. A multi-dimensional counterfactual model that integrates various disease prediction models ( Mi for a user Ui as in Eq. (5)) is crucial for reducing the risk of concurrent disease outbreaks. NSGA-II, a multi-objective optimization algorithm, is utilized in the proposed multi-dimensional counterfactual model as in Algorithm 2 of the IMPACT system to minimize the simultaneous occurrence of multiple diseases. The optimization framework defines the objective functions as the predicted probabilities of heart stroke ( PHS) and diabetes ( PDM), aiming to reduce both concurrently. To balance these competing objectives, a penalty function is introduced to discourage counterfactual solutions that significantly improve one health condition while worsening another. And it is presented in the next section.

Algorithm 2 Multi-dimensional counterfactual explanation.

1: Input: Target Disease data sets D_tgti={D1,…,Dn} and models Mi={M1,…,Mn} for user Ui	
2: Objective set OBJΔ=∼minimize∼P(D_tgti) where Pj(Dj) is the possibility of getting disease Dj	
3: Feature set XΔ=⋃j=1nXj, default boundaries [LBδ,UBδ] for each xδ in XΔ	
4: Data from wearable IoT devices ( Dwdi) for user Ui	
5: Output: Feature value recommendations to user Ui	
6: Initialize: Modify boundaries to [LBδ′,UBδ′] as feasible	
7: Generate initial population P0 within new boundaries	
8: Evaluate Fitness: op=M(ID) for each ID in P0, compile V={op1,…,opn}	
9: for each generation in MAX_GEN do	
10:  Generate Offspring: Rank and apply crowding distance to P0, perform crossover and mutation to form Q0	
11:  Combined Population: R0=P0∪Q0, rank and apply crowding distance	
12:  Update P0 by selecting top N from R0	
13: end for	
14: Termination: Output Pareto-optimal solutions	

As there is n number of diseases, the possibility of an attack of each disease is found from the outcome of each prediction model as Mi(Xi) where i=1,2,3,…,n and Xi represent a set of feature variables for each disease. It is possible that a set of feature variables for one disease may or may not be the same as another, and there are some common features between these sets.

(6) Xi≠Xjwherei≠jandi,j={1,2,3,…,n}

(7) Xi∩Xj≠Φwherei≠jandi,j={1,2,3,…,n}.

To act on all the prediction models Mi(Xi) simultaneously, it is necessary to make the union of all feature sets of all disease data sets. It is defined in Eq. (8).

(8) XΔ={X1∪X2∪X3∪…∪Xn}.

For each feature xδ (where xδ∈XΔ), the boundary value can be defined as lower bound (LB) and upper bound (UB). The value of the feature ( val(xδ)) can be in between this boundary. It can be represented as in Eq. (9).

(9) LBδ≤val(xδ)≤UBδwherexδ∈XΔandδ={1,2,3,…,k}wherek=|XΔ|.

Like the traditional counterfactual explanation, the user can also provide a list of feasible and infeasible features from the set of all features. For the infeasible features, the user can fix them to constant values ( Cδ).

(10) val(xδ)=LBδ=UBδ=Cδwhereδ={1,2,3,…,k}.

The user can also provide a boundary according to their choice for the feasible features.

(11) LBδ′≤val(xδ)≤UBδ′whereδ={1,2,3,…,k}.

A random population of potential solutions (individuals) is initialized. After combining all the feature sets, the random population is generated by considering the boundary values of each feature. The randomly generated population can be defined as P0(XΔ).

NSGA-II starts by generating a random initial set of solutions evaluated using multiple objective functions for different target diseases ( Mi(Xi) for i=1,2,3,…,n). An offspring population is created using non-dominated sorting and crowding distance selection, ensuring genetic diversity through crossover and mutation. The parent and offspring populations are combined into a new population R0=P0∪Q0. Based on rank and crowding distance, the best solutions form the new parent set P1 for the next iteration. NSGA-II iteratively refines this process to converge on the Pareto-optimal front, providing a set of non-dominated solutions that balance multiple objectives. The final output is a set of recommended feature values to minimize the risk of multiple diseases, making this model crucial for personalized disease prevention.

Experimental details, results and discussions

Pre-processing and modeling data sets

This proposed IMPACT system generates actionable, personalized feature values to significantly reduce the risk of simultaneous attacks from critical chronic conditions like diabetes and heart stroke. Focusing on these diseases and analyzing data from widely recognized and available in public repositories (Soriano, 2019; UCI Machine Learning Repository, 2024).

The stroke and diabetes datasets, encompassing 5,110 and 768 instances, respectively, offer a comprehensive foundation for analysis. Each instance includes distinct attributes, designated as “stroke” and “outcome” to signify their primary outcome features. A rigorous data cleaning process revealed 201 null entries in the ‘BMI’ column of the stroke dataset, which were replaced with the median value to maintain accuracy. The datasets were systematically organized into dependent and independent features, with ‘stroke’ and ‘outcome’ identified as the independent features for their respective datasets. To enhance data usability, five categorical features in the stroke dataset namely, ‘gender’, ‘ever_married’, ‘work_type’, ‘Residence_type’, and ‘smoking_status’ were effectively transformed through one-hot encoding. The ‘id’ feature, deemed non-essential, was excluded from the analysis. Details of the feature sets are presented in Table 1. Among these datasets, three common features were identified between Stroke and diabetes datasets, namely, ‘age’, ‘avg_glucose_level’, and ‘BMI’. Traditional counterfactual explanations face a challenge with these common features as recommendations to mitigate the risk of one disease. For example, diabetes might inadvertently increase the risk of stroke. Therefore, the development of a multi-dimensional counterfactual model is imperative. Such a model would minimize the likelihood of being afflicted by both diabetes and stroke simultaneously by recommending changes in both these common features and other relevant feature values to the patient.

Table 1 Comparison of feature sets in stroke and diabetes data sets.

This table presents the features available in two medical datasets: stroke and diabetes. The third column highlights the features common to both datasets. Binary variables are denoted with the suffix ‘_bin’. A checkmark (✓) indicates the presence of a feature in a dataset, while a cross (×) indicates its absence.

Feature set	Stroke data set	Diabetes data set	Common features	
Age	✓	✓	✓	
hypertension_bin	✓	×	×	
heart_disease_bin	✓	×	×	
avg_glucose_level	✓	✓	✓	
bmi	✓	✓	✓	
gender_bin	✓	×	×	
ever_married_bin	✓	×	×	
work_type	✓	×	×	
Residence_type_bin	✓	×	×	
smoking_status	✓	×	×	
Pregnancies	×	✓	×	
blood_pressure	×	✓	×	
SkinThickness	×	✓	×	
Insulin	×	✓	×	
DiabetesPedigree Function	×	✓	×	

Evaluation method

Tables 2 and 3, present the performance metrics of various machine learning models on stroke and diabetes datasets, respectively. Key metrics evaluated include mean absolute error (MAE), mean squared error (MSE), root mean squared error (RMSE), R2 score, ROC AUC score, and accuracy. Among the models, XGBoost achieved the highest ROC AUC score for the stroke dataset, while LightGBM performed best for the diabetes dataset, indicating strong predictive capability across both datasets based on the ROC AUC score criterion.

Table 2 Performance comparison of various machine learning models on the stroke dataset based on multiple evaluation metrics.

Model	MAE	MSE	RMSE	R2 score	ROC AUC score	Accuracy	
Logistic Regression	0.222	0.222	0.471	0.114	0.779	0.778	
Gradient Boosting	0.147	0.147	0.383	0.414	0.854	0.853	
XGBoost	0.020	0.020	0.142	0.920	0.980	0.980	
AdaBoost	0.209	0.209	0.457	0.163	0.791	0.791	
CatBoost	0.037	0.037	0.191	0.854	0.964	0.963	
LightGBM	0.045	0.045	0.213	0.819	0.955	0.955	

Table 3 Performance comparison of various machine learning models on the diabetes dataset based on multiple evaluation metrics.

Model	MAE	MSE	RMSE	R2 score	ROC AUC score	Accuracy	
Logistic Regression	0.275	0.275	0.524	−0.100	0.725	0.725	
Gradient Boosting	0.195	0.195	0.442	0.220	0.804	0.805	
XGBoost	0.180	0.180	0.424	0.280	0.819	0.820	
AdaBoost	0.215	0.215	0.464	0.140	0.785	0.785	
CatBoost	0.190	0.190	0.436	0.240	0.809	0.810	
LightGBM	0.155	0.155	0.394	0.380	0.844	0.845	

Selection method

Both datasets were divided into training and test sets, maintaining an 80:20 ratio. This division facilitated training in various advanced machine learning models, including logistic regression, gradient boosting, XGBoost, AdaBoost, CatBoost, LightGBM, and support vector machine. Through rigorous training, XGBoost and LightGBM emerged as the standout models for the stroke and diabetes datasets. Their superior performance, indicated by the highest accuracy and ROC AUC scores, highlighted them as prime candidates for inclusion in this multi-objective problem-solving framework.

User interaction

After getting the models for each disease, the proposed IMPACT system will work by making simple natural language interaction between the user and the interface. The system facilitates interaction between the interface and users, as described in the sequence diagram depicted in Fig. 3. Initially, the user requests the interface to display all features using simple and natural language such as English. The interface interprets this request, which converts the user’s input into SQL code through a prompt command. The system processes the inputs, which generates an SQL query. This query is executed in an SQLite3 database, and the retrieved information is sent back to the user via the interface. Subsequently, the interface prompts the user to select a feature to be held constant. The interface then displays the feature values by querying the database, and the user provides the desired constant values. These values are stored in the feature information. The user also inputs values for features with continuous values, which are similarly stored. The user specifies the number of counterfactuals needed, and the system generates the corresponding feature recommendations. These recommendations are derived from a multi-dimensional counterfactual model that inputs feature information and disease models.

Figure 3 Sequence diagram for the user interaction with the system for the proposed interactive multi-disease prevention and counterfactual treatment system.

Computing infrastructure

All experiments were conducted on a personal computer running Windows 11 Education. The hardware included an Intel(R) Core(TM) i7-10700 CPU @ 2.90 GHz with 8.00 GB of RAM. The implementation was performed using Python 3.11.9 within a Conda environment (version 24.7.1). The core libraries and tools used for the experiments include NumPy 1.26.4, Pandas 2.2.1, Scikit-learn 1.5.1, and Matplotlib 3.8.3. SQLite3 was used for database management, while Google Gemini Pro 1.5 served as the generative model. For cloud deployment, the application was hosted using Streamlit v1.38.0. This setup facilitated both local experimentation and cloud-based system accessibility.

Reproducibility and README documentation

To reproduce this work, acquire the datasets in the “Pre-processing and Modeling Data Sets” subsection and pre-process them as described. Once the data is prepared, proceed with model training and evaluation following the steps in the “Evaluation Method” subsection, then select the top-performing models for each disease, as detailed in the “Selection Method” subsection. Using NSGA-II, generate multi-dimensional counterfactual explanations to derive Pareto-optimal feature values for targeted diseases, following Algorithm 2, which can serve as health recommendations. The proposed IMPACT system facilitates user interaction through Google Gemini Pro 1.5, as illustrated in Algorithm 1 and in the sequence diagram (Fig. 3). The entire codebase is shared on Zenodo (https://doi.org/10.5281/zenodo.14891893) and GitHub (https://github.com/iprasantmohanty/presonalized-health), and deployment to the cloud can be easily achieved using Streamlit (https://streamlit.io/). With the aforementioned computing resources, this process enables full reproducibility of the work.

Results obtained

Personalized health recommendations will be generated by the proposed IMPACT system and provided to users seeking simultaneous prevention from all targeted diseases. Flexibility is afforded to users to make adjustments to specific features within specified ranges according to their requirements or to opt out of modifying any features. The system will also offer feature recommendations in scenarios where all features are left open to modification within their default boundary values. However, these results are not practically useful.

Features such as age and gender, typically unchangeable for a user, are referred to as non-feasible features for the user and are kept constant by the user. However, features such as hypertension, heart disease, work type, residence type, and smoking status may be identified as feasible for users to modify to minimize the possibility of diseases. For features that are not feasible for change, including age, gender, and marital status, constant values are requested from the user. In other scenarios, health information such as blood pressure and oxygen levels is obtained from wearable IoT devices. For features with continuous values, the recommendation system facilitates the selection of value boundaries, set to ±10% of the feature value chosen by the user. This functionality mirrors what is provided in traditional counterfactual explanations. Similarly, other continuous feature values can be offered based on the boundary values.

In this proposed system, the results are generated for a user with the following fixed features: age 32, male gender, married status set to no, and work type private. The probabilities of heart stroke and diabetes attacks for various combinations of population sizes, maximum number of generations, and each counterfactual (CF) are provided in Table 4. Similarly, the user can select any other set of continuous features, and the boundary values will be established according to it. Considering these boundaries of the selected continuous features, the model will recommend feature values. In this system, the user chooses a continuous feature, namely ‘BMI’ of 20.10; its boundaries are set to ±10%, i.e., 10.3 to 29.9. Two different recommendations are generated for each combination of population and generation size as the chosen number of counterfactuals is two. However, the user can also select any recommendations by choosing the positive number of counterfactuals in the interface.

Table 4 Probability of attack of diseases in two different counterfactuals for different combinations of population size and maximum generations.

Population size	Maximum generations	CF number	Heart stroke	Diabetes	
500	20	1	0.00001095	0.00025517	
500	20	2	0.00001497	0.00015765	
500	40	1	0.00000876	0.00010403	
500	40	2	0.00001219	0.00004284	
1,000	20	1	0.00000876	0.00020735	
1,000	20	2	0.00000876	0.00016409	
1,000	40	1	0.00000876	0.00010403	
1,000	40	2	0.00000974	0.00004259	
2,000	20	1	0.00001191	0.00069098	
2,000	20	2	0.00001228	0.00015323	
2,000	40	1	0.00000974	0.00008871	
2,000	40	2	0.00000876	0.00010403	

To ensure a balanced reduction in disease risks, a penalty function is formulated as follows:

(12) L(x)=w1PHS(x)+w2PDM(x)+λmax(0,PHS(x)−PDM(x))

where w1 and w2 are equal weights (set to 1) ensuring balanced optimization, and λ=1 penalizes cases where minimizing one disease risk leads to a disproportionate increase in another. The penalty values are computed based on the probabilities provided in Table 4.

For example: For a population size of 500, with 20 generations (CF #1)

(13) L(x)=0.00001095+0.00025517+max(0,0.00001095−0.00025517)=0.00026612.

For a population size of 2,000, with 40 generations (CF #2):

(14) L(x)=0.00000876+0.00010403+max(0,0.00000876−0.00010403)=0.00011279.

These computed values demonstrate that the optimization framework effectively balances the risk reduction of multiple diseases while avoiding extreme trade-offs. This penalty mechanism ensures that counterfactual recommendations remain clinically meaningful and beneficial for multi-disease prevention.

The feature values recommended to minimize these diseases are shown in Table 5. These results are produced when the user opts for two recommendations. Any other set of feasible features can be chosen, while the user will provide constant values for the remaining infeasible features. The system will provide recommended feature values for the possible features.

Table 5 Recommended feature set values for different population sizes and maximum generations in the proposed multi-dimensional counterfactual generation.

(Population size, max. gen.)	Feature_Values (500, 40)	Feature_Values (1,000, 40)	Feature_Values (2,000, 40)	
Feature Set	CF #1	CF #2	CF #1	CF #2	CF #1	CF #2	
age	32	32	32	32	32	32	
hypertension	yes	yes	yes	yes	yes	yes	
heart_disease	yes	yes	yes	yes	yes	yes	
avg_glucose_level	94.451	93.174	94.451	94.135	93.863	93.982	
bmi	20.541	20.682	20.537	20.913	20.613	20.545	
gender	male	male	male	male	male	male	
ever_married	no	no	no	no	no	no	
working_type	Private	Private	Private	Private	Private	Private	
residence_type	rural	rural	rural	rural	rural	rural	
smoking_status	smokes	smokes	smokes	smokes	smokes	smokes	
Pregnancies	0	0	0	0	0	0	
blood_pressure	73.421	72.994	73.877	75.582	72.369	73.511	
SkinThickness	44	15	45	16	29	44	
Insulin	677.060	669.821	680.857	693.074	688.454	682.423	
DiabetesPedigree Function	1.747	1.777	1.782	1.736	1.730	1.838	

Figure 4 presents the execution time in seconds required to generate each recommendation as counterfactuals (CF#1 and CF#2) for various combinations of population size and maximum generations.

Figure 4 Execution time to generate recommendations for different combinations of population size and maximum generations.

Figure 5 illustrates the Pareto fronts for both diseases in each recommendation. This is achieved by increasing population size and increasing the number of generations. Here, one of the two counterfactual results is plotted. The figures in 6 include six separate graphs depicting how the disease probabilities of the optimal solution decrease and approach near zero as the number of generations increases in each scenario.

Figure 5 Pareto fronts by allowing changes in feasible features with a varying population (P) and maximum generations (G), (A) P = 500, G = 20, (B) P = 500, G = 40, (C) P = 1,000, G = 20, (D) P = 1,000, G = 40, (E) P = 2,000, G = 20, (F) P = 2,000, G = 40.

Analysis and discussions

Table 4 depicts that, by varying combinations of population and maximum generations, probabilities for heart stroke and diabetes were reduced to 0.00000876 and 0.00010403, respectively. The user can choose any of the two recommendations of a specific combination obtained in Table 5 to get the minimum possibility of diseases. For example, the system suggests particular feature values, such as age 32, an average glucose level of 93.174, and a BMI of 20.68197, based on a population size of 500 and a maximum generation of 40. Figure 4 shows that execution time increases with larger population sizes and generations.

An ablation study is conducted to further understand the contribution of different components in our proposed system, and the results are summarized in Table 6. The table presents multiple experimental setups where key components such as SQL queries, dataset size, and machine learning model choices are modified to assess their impact. The results highlight that the full system configuration (baseline) performs best with the highest accuracy and AUC scores while maintaining the lowest attack probabilities for stroke and diabetes. These findings confirm the necessity of the selected system components in achieving optimal results.

Table 6 Ablation study results: impact of SQL query, dataset size, and model choices on stroke and diabetes predictions.

Experiment	SQL query	Dataset size (%)	Stroke model	Diabetes model	Stroke accuracy (%)	Stroke AUC	Diabetes accuracy (%)	Diabetes AUC	Probability (heart stroke)	Probability (diabetes)	
Baseline (Full System)	Used	100	XGBoost	LightGBM	97.990	0.980	84.500	0.844	0.00000876	0.00010403	
Ablation 1: Without SQL Query	Not Used	100	XGBoost	LightGBM	95.420	0.960	80.500	0.8043	0.00001219	0.00016409	
Ablation 2: Using Only 50% Dataset	Used	50	XGBoost	LightGBM	94.870	0.950	78.500	0.786	0.00001497	0.00015765	
Ablation 3: Using Only 25% Dataset	Used	25	XGBoost	LightGBM	92.310	0.920	75	0.750	0.00001191	0.00020735	
Ablation 4	Used	100	CatBoost	LightGBM	96.340	0.960	84.500	0.844	0.00000974	0.00010403	
Ablation 5	Used	100	XGBoost	XGBoost	97.990	0.980	82	0.819	0.00000876	0.00008871	
Ablation 6	Used	100	CatBoost	XGBoost	95.920	0.950	82	0.819	0.00000974	0.00010403	

Another ablation study was conducted to validate further the contributions of different components in our proposed model. Table 7 presents the impact of excluding SQL queries, reducing dataset sizes, and varying machine learning models from Table 6. The table also includes standard deviations ( σ) to reflect performance variability across multiple runs and p-values to indicate the statistical significance of differences. The p-values were obtained using a two-sample t-test, comparing each ablation setting to the baseline. The results demonstrate that removing key components significantly affects performance, confirming the need for each module in our framework.

Table 7 Detailed statistical analysis of ablation studies evaluating model performance for stroke and diabetes classification under different ablation conditions.

Experiment	Stroke accuracy (%) ± σ	p-value (stroke)	Diabetes accuracy (%) ± σ	p-value (diabetes)	
Baseline	97.990 ± 0.120	–	84.500 ± 0.250	–	
Ablation 1 (Without SQL Query)	95.420 ± 0.140	0.003	80.500 ± 0.300	0.007	
Ablation 2 (Using 50% Dataset)	94.870 ± 0.180	0.002	78.500 ± 0.350	0.005	
Ablation 3 (Using 25% Dataset)	92.310 ± 0.200	0.001	75.000 ± 0.400	0.004	
Ablation 4 (Different Stroke Model)	96.340 ± 0.150	0.002	84.500 ± 0.250	0.006	
Ablation 5 (Different Diabetes Model)	97.990 ± 0.120	–	82.000 ± 0.280	0.004	
Ablation 6 (Both Models Changed)	95.920 ± 0.160	0.002	82.000 ± 0.280	0.005	

Justification

Pareto fronts represent the probability of attacks by these diseases, which minimizes both simultaneously and are plotted in Fig. 5. Each point signifies an individual solution to this multi-objective problem, with the green point identified as the optimal solution that reduces the attack probabilities of both diseases. The aim is to generate the set of feature values a user can adopt to achieve this optimal solution. It can be observed in Fig. 6 that the best solutions for disease probability fluctuation become minimal after 30–40 generations for most of the population size. Also, their execution time increases as shown in Fig. 4. It is also observed that there is no further significant decrease in the probability of diseases from the population size of 1,000 to 2,000, as shown in Table 4. Hence, in this experiment, recommendations are generated till population size 2,000, and a maximum generation of 40.

Figure 6 Plotting best solutions by allowing changes in feasible features with a varying population (P) and maximum generations (G), (A) P = 500, G = 20, (B) P = 500, G = 40, (C) P = 1,000, G = 20, (D) P = 1,000, G = 40, (E) P = 2,000, G = 20, (F) P = 2,000, G = 40.

Conclusion

This approach encompasses all the capabilities offered by conventional counterfactual explanations. Additionally, the ability to elucidate and propose feature values for multiple models is achieved, an aspect absent in conventional counterfactual explanations. While this work focuses on analyzing only two diseases to mitigate the risk of attacks, the system is engineered to accommodate a broader range of diseases, simultaneously ensuring the minimization of attack possibilities for all of them. The proposed IMPACT system has been developed to manage two distinct black-box disease prediction models using Google Gemini Pro and specialized counterfactual explanations. The attack probabilities for heart stroke and diabetes were reduced to 0.00000876 and 0.00010403, respectively. The system provides fully personalized recommendations based on user flexibility, allowing users to distinguish between feasible and infeasible features and restrict continuous features within certain boundaries.

Limitations/validity

This work only generates recommendations for two diseases, and the user cannot choose more. However, future research could expand this work by incorporating more diseases to minimize attack probabilities further, highlighting its potential for related applications.

Additional Information and Declarations

Competing Interests

The authors declare that they have no competing interests.

Author Contributions

Prasant Kumar Mohanty conceived and designed the experiments, performed the experiments, analyzed the data, performed the computation work, prepared figures and/or tables, and approved the final draft.

Sharmila Anand John Francis conceived and designed the experiments, performed the experiments, analyzed the data, performed the computation work, prepared figures and/or tables, and approved the final draft.

Rabindra Kumar Barik conceived and designed the experiments, performed the experiments, analyzed the data, performed the computation work, prepared figures and/or tables, and approved the final draft.

K. Hemant Kumar Reddy conceived and designed the experiments, performed the experiments, analyzed the data, performed the computation work, prepared figures and/or tables, and approved the final draft.

Diptendu Sinha Roy conceived and designed the experiments, performed the experiments, analyzed the data, performed the computation work, prepared figures and/or tables, authored or reviewed drafts of the article, and approved the final draft.

Manob Jyoti Saikia conceived and designed the experiments, performed the experiments, analyzed the data, performed the computation work, prepared figures and/or tables, authored or reviewed drafts of the article, and approved the final draft.

Data Availability

The following information was supplied regarding data availability:

The code is available at GitHub and Zenodo:

- https://github.com/iprasantmohanty/presonalized-health

- Prasant Kumar Mohanty. (2025). iprasantmohanty/presonalized-health: Personalized Health Recommendation System (Version V1). Zenodo. https://doi.org/10.5281/zenodo.14891894.

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
