# Peer review of "IMPACT: an interactive multi-disease prevention and counterfactual treatment system using explainable AI and a multimodal LLM"

_PeerJ Computer Science, doi:10.7717/peerj-cs.2839_

## Round 0.1 · original submission · Major Revisions

This manuscript addresses a highly relevant and timely topic: using AI for multi-disease prediction and personalized recommendations in healthcare. The concept of a system like IMPACT has significant potential. However, the manuscript, in its current form, suffers from critical methodological gaps and a lack of clarity in its presentation, particularly regarding the ML architecture, data integration, and evaluation. Therefore, a major revision is required.

The most important issues in the current manuscript involve insufficient detail on how the diverse data sources (smartwatches, fitness trackers, biosensors) are integrated and lack of essential statistical measures (p-values, standard deviations).

Addressing all the points raised by the reviewers, as well as the additional points noted above, is essential for the manuscript to be reconsidered for publication. The lack of detail regarding the core methodology and the insufficient evaluation are the most critical issues that need to be addressed.

Reviewer 1 ·

Basic reporting

(CS105833)

I would like to thank the authors for submitting this great work in PeerJ-CS. While the study covers a significant topic on counterfactual explanations and personalized recommendations in healthcare prediction models, the manuscript requires major revisions before it can be considered for publication. My main concerns are regarding the ML structure that was used. Please find my detailed comments below:

IMPACT: An interactive multi-disease prevention and counterfactual treatment system using explainable AI and a multimodal LLM


Introduction
Line 55 - Research gap: The introduction does not clearly articulate the specific problem that XAI-IMPACT aims to address. To enhance clarity and impact, this section should explicitly define the key challenges in current ML techniques, such as deployment difficulties on resource-constrained devices, the trade-offs between speed and accuracy, and the limitations in handling small datasets. By outlining these specific obstacles, the authors can better justify the need for a novel model like XAI-IMPACT to effectively manage multi-disease scenarios.

Methodology:
Line 195: Similar to Figure 1, I recommend that the authors include an additional diagram illustrating the structure of their ML model and the workflow of the NSGA-II optimization algorithm. Currently, the system feels somewhat like a black box, and incorporating these visual representations would greatly enhance clarity and provide the audience with a better understanding of the mechanisms behind XAI-IMPACT. This addition would effectively demystify the "magic" of the system and improve the overall comprehensibility of the manuscript.

Line 204: The description of the data used in the study appears overly general. The authors mention employing datasets from sources such as smartwatches, fitness trackers, and biosensors. However, there is no detailed explanation of how these diverse data streams are integrated into a single ML algorithm. Could the authors provide more clarity on the following points?
Data Coupling: How are these heterogeneous data types (e.g., time-series data from smartwatches, discrete measurements from biosensors) harmonized and preprocessed to be compatible with a single ML algorithm? Is there a specific data fusion or feature engineering technique employed?
Data Dimensions: Could the authors elaborate on the dimensions of the data? For example, how many features are there, what is the frequency or resolution of the data streams, and how is missing or noisy data handled?
Model Design: How does the ML algorithm accommodate these varied data formats? Are there specific architectural adjustments or techniques (e.g., multi-input models, embedding layers) to ensure effective handling of different data types?
Providing a detailed description of these aspects would significantly strengthen the manuscript by demonstrating the robustness and scalability of the proposed approach.

Line 248: The authors mentioned that they used NSGA-II to minimize the possibility of multiple diseases attacking simultaneously. Can the authors comment on the mathematical cost function of this multi-objective optimization algorithm that was used? How did the authors use a penalty function in their model? It would be beneficial to the manuscript to mention these equations, along with all other equations mentioned to describe the framework.

Line258: The authors discussed the various features of the massive datasets utilized in their study. However, a common challenge faced when working with such diverse datasets is the presence of missing data for certain features. To address this issue, various data imputation or missing data handling techniques are commonly employed. Could the authors elaborate on how their framework addresses this issue, if at all? The authors can take a look at the review paper with similar problem: 2024 Sharifi et al. - Application of artificial intelligence in digital twin models for stormwater infrastructure systems in smart cities


Line 309: The authors employed several ML algorithms, including Logistic Regression (LR), XGBoost, and others, and reported that LightGBM outperformed the rest based on various statistical indices. However, the manuscript does not provide a discussion on why LightGBM demonstrated superior performance. To improve the depth of the analysis, the authors could consider discussing potential reasons for this observation (feature Interactions - data characteristics)

Results
Line 377: The resolution for figure 3 is too low for publication quality in this journal. I suggest the authors plot this figure with higher resolution.

Discussion and Conclusion
Line 385: This part seems to be too short. I would suggest adding more fruitful discussions on issues such as the scalability of the IMPACT system to manage a broader range of diseases, the computational efficiency when handling multiple models, and the potential trade-offs between minimizing attack probabilities and maintaining model interpretability. Additionally, a deeper exploration of the challenges and opportunities in real-world implementation, such as user adoption and integration with existing healthcare systems, would significantly enrich the discussion.

Also, the authors briefly mentioned their major findings, highlighting the novelty and adaptability of the proposed IMPACT system in providing personalized counterfactual explanations and minimizing attack probabilities across multiple black-box disease prediction models. However, the limitations of this framework are not mentioned, and there is no discussion of future work. Including a brief mention of limitations and directions for future research would provide a more well-rounded conclusion.

Experimental design

Please find all the comments on the first part.

Validity of the findings

Please find all the comments on the first part.

Additional comments

Please find all the comments on the first part.

Reviewer 2 ·

Basic reporting

The motivation for using recommendations is not clearly articulated. The introduction should provide a more thorough discussion of the reasons behind their use.

The techniques employed for generating recommendations are unclear. Based on Figure 1, it appears that Retrieval-Augmented Generation (RAG) has been utilized; however, RAG is not discussed in the paper. The paper should explicitly clarify which techniques have been applied.

Experimental design

Ablation studies should be included to better understand the contribution of each component, such as the SQL query, prompts, dataset sizes, and so on.

Validity of the findings

No p-values or standard deviations are reported, making it difficult to assess the statistical significance of the results.

---

## Round 0.2 · Minor Revisions

Dear Authors,

I am pleased to inform you that the current version of your manuscript has significantly improved compared to the previous submission. The reviewers are satisfied with the revisions you have made. However, some minor adjustments to the manuscript's form are still required.

Specifically, I have the following comments:

- Decimal Places in Tables: Please standardize the number of significant decimal places used throughout all tables in the manuscript. Ensure consistency in the presentation of numerical data.
- Figure and Table Captions: The current figure and table captions are too brief and require more detailed descriptions of their content. Please expand the captions with additional information and provide clear references to the relevant text within the manuscript.
- Table 2 Format: While Table 2 appears to address the reviewers' concerns, please double-check that it strictly adheres to the journal's required formatting guidelines.
- Figure 3 Font: The font used in Figure 3 is generally adequate. However, due to the figure's size, the text becomes difficult to read. Please adjust the font size or style to improve readability.
Please address these minor revisions and resubmit your revised manuscript. We look forward to receiving your updated submission.

Sincerely,

Marta Lovino

Reviewer 1 ·

Basic reporting

The authors have addressed my comments; therefore, the paper can be accepted for publication in the present format.

Experimental design

The authors have addressed my comments; therefore, the paper can be accepted for publication in the present format.

Validity of the findings

The authors have addressed my comments; therefore, the paper can be accepted for publication in the present format.

Reviewer 2 ·

Basic reporting

The authors have addressed may concerns, so I recommend acceptance.

Experimental design

The authors have addressed may concerns, so I recommend acceptance.

Validity of the findings

The authors have addressed may concerns, so I recommend acceptance.

---

## Round 0.3 · accepted · Accept

I am writing to recommend the acceptance of the Manuscript.
The manuscript is well-written, clearly structured, and makes a valuable contribution to the field.

The authors have effectively addressed all concerns raised during the review process. Therefore, I recommend that the manuscript be accepted for publication.